# Flow decorrelation in heavy-ion collisions at $\sqrt{s_{\mathrm{NN}}} = 27$ and 200 GeV with 3D event-by-event viscous hydrodynamics

J. Cimerman[1*], Iu. Karpenko[1], B. Tomášik[1,2] and B. A. Trzeciak[1]

**1** Faculty of Nuclear Sciences and Physical Engineering, Czech Technical University in Prague, Prague, Czech Republic
**2** Univerzita Mateja Bela, Banská Bystrica, Slovakia
* jakub.cimerman@fjfi.cvut.cz

February 8, 2022

## Abstract

**We present the first calculation of longitudinal decorrelation of anisotropic flow at RHIC Beam Energy Scan (BES) energies using event-by-event viscous hydrodynamic model (vHLLE), with two different initial states (GLISSANDO2 and UrQMD) and hadronic cascade. We investigate the origin of the observed decorrelation by checking separately flow angle and flow magnitude decorrelation and by calculating decorrelation in the initial state eccentricity.**

## 1   Introduction

Research of the anisotropic flows is quite popular in heavy-ion community, since it can reveal new information about properties of quark-gluon plasma. During last few years this research started to focus also on the longitudinal dependence of the anisotropic flows. First experimental results on longitudinal decorrelation were obtained at LHC energies by CMS and ATLAS [1–3], while at RHIC energies there are only preliminary results, yet [4, 5]. This paper sums up the first calculation of anisotropic flow decorrelation using hydrodynamic model at RHIC BES energies [6].

## 2   Model

The results presented in this work have been obtained with an event-by-event hybrid model consisting of four parts. First, initial conditions are generated either using Monte Carlo Glauber model GLISSANDO2 [7] extended to 3D, or with UrQMD model [8], which simulates nucleon-nucleon scatterings through string formation and subsequent string break-up. Next, the evolution of the hot and dense state is simulated using viscous hydrodynamic code vHLLE [9]. A Monte Carlo hadron sampling is performed according to Cooper-Frye formula [10] and final-state rescatterings and resonance decays are simulated using UrQMD cascade. Moreover, the model includes a

finite baryon and electric charge density at all stages. More details about the model can be found in [11].

## 3   Results

From the symmetry of the collision one might expect, that the anisotropic flow would be invariant along longitudinal direction. However, the event-by-event fluctuations can violate this symmetry. To get the quantitative measure of the change of anisotropic flow along the longitudinal direction, we calculate the factorization ratio defined as:

$$r_n(\eta, \eta_{\mathrm{ref}}) = \frac{\langle \mathbf{V}_n(-\eta)\mathbf{V}_n^*(\eta_{\mathrm{ref}})\rangle}{\langle \mathbf{V}_n(+\eta)\mathbf{V}_n^*(\eta_{\mathrm{ref}})\rangle} = \frac{\langle v_n(-\eta)v_n(\eta_{\mathrm{ref}})\cos n(\Psi_n(-\eta) - \Psi_n(\eta_{\mathrm{ref}}))\rangle}{\langle v_n(+\eta)v_n(\eta_{\mathrm{ref}})\cos n(\Psi_n(+\eta) - \Psi_n(\eta_{\mathrm{ref}}))\rangle}, \qquad (1)$$

where $\mathbf{V}_n = v_n e^{in\Psi_n}$ is the flow vector. If the ratio reaches 1, the anisotropic flows at $+\eta$ and $-\eta$ are equal. However, when it goes below 1, the flows are more and more different.

When we look closer at Eq. (1) we may notice, that this correlator may be split into two:

$$r_n^v(\eta) = \frac{\langle v_n(-\eta)v_n(\eta_{\mathrm{ref}})\rangle}{\langle v_n(+\eta)v_n(\eta_{\mathrm{ref}})\rangle}, \qquad (2a)$$

$$r_n^{\Psi}(\eta) = \frac{\langle \cos n(\Psi_n(-\eta) - \Psi_n(\eta_{\mathrm{ref}}))\rangle}{\langle \cos n(\Psi_n(+\eta) - \Psi_n(\eta_{\mathrm{ref}}))\rangle}. \qquad (2b)$$

Now we have separated factorisation ratio into parts responsible for flow magnitude and flow angle. Thanks to this we can investigate, where does the decorrelation originate from.

The factorisation ratio as a function of pseudorapidity compared with STAR preliminary data is shown in Figure 1. While calculations with GLISSANDO initial state can reproduce the data within uncertainties, UrQMD initial state overestimate the decorrelation.

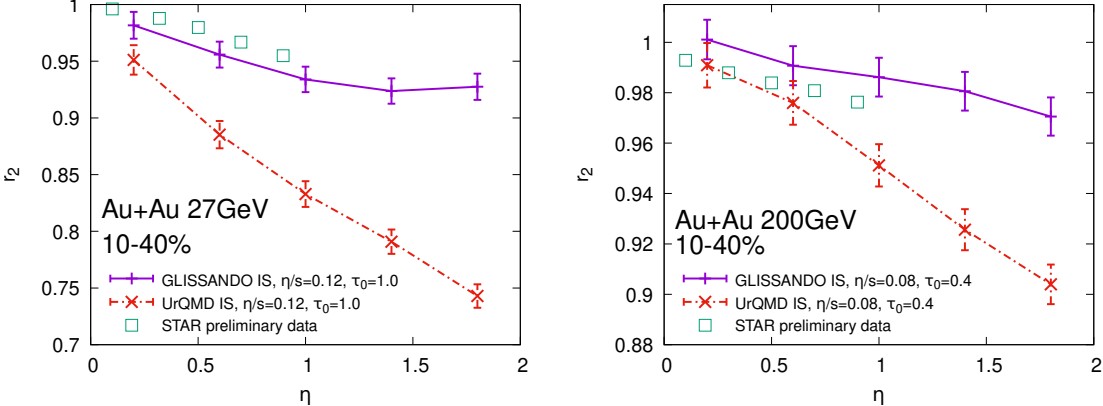

Figure 1: Pseudorapidity dependence of the factorization ratio $r_2$ for $10 - 40\%$ Au-Au collisions at $\sqrt{s_{\mathrm{NN}}} = 27$ (left) and 200 GeV (right) compared with STAR preliminary data [4,5]. Figure taken with permission from [6].

In Figure 2 we show the two factorisation ratios from Eqs. (2). These calculations show, that the flow angle decorrelation surpasses the flow magnitude decorrelation independent of the initial state model. The same conclusion has been found at LHC energies [12].

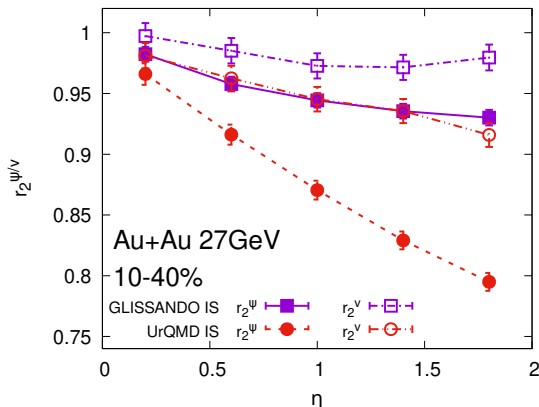

Figure 2: Pseudorapidity dependence of the decorrelation of the flow magnitude $r_2^v$ and the flow angle $r_2^\Psi$ of charged hadrons for $10-40\%$ Au-Au collisions at $\sqrt{s_{\mathrm{NN}}} = 27$ GeV. Figure taken with permission from [6].

To understand the big difference between the initial state models, we need to investigate initial state spatial eccentricities, since they are strongly correlated with the anisotropic flows. Therefore, we defined the factorisation ratio of initial-space eccentricities:

$$r_n^\epsilon(\eta_s) = \frac{\left\langle \epsilon_n(-\eta_s)\epsilon_n(\eta_{s,\mathrm{ref}}) \cos[n\left(\Psi_n(-\eta_s) - \Psi_n(\eta_{s,\mathrm{ref}})\right)]\right\rangle}{\left\langle \epsilon_n(\eta_s)\epsilon_n(\eta_{s,\mathrm{ref}}) \cos[n\left(\Psi_n(\eta_s) - \Psi_n(\eta_{s,\mathrm{ref}})\right)]\right\rangle}, \tag{3}$$

where

$$\epsilon_n e^{in\Psi_n} = \frac{\int e^{in\phi} r^n \rho(\vec{r}) d\phi\, r\, dr}{\int r^n \rho(\vec{r}) d\phi\, r\, dr}.$$

Figure 3 shows this factorisation ratio as a function of space-time rapidity. When we compare this figure to Figure 1, we may notice that these two factorisation ratios agree even quantitatively. This means that the longitudinal decorrelation originates already from the initial state, where the difference between models is caused by different physical principle they use, and also by different parametrization of switching from hadron cascade to fluid, which is discussed in more detail in [6].

## 4   Conclusion

This paper summarizes the results of the first calculation of longitudinal decorrelation of elliptic flow with a hydrodynamic model at RHIC BES energies. We showed that calculation with the GLISSANDO initial state describes preliminary data quite well, while the UrQMD initial state results in overestimated decorrelation. We found that this difference originates from the initial state and can be seen in decorrelation of initial-state eccentricity. Moreover, we showed that the flow angle has bigger contribution to total decorrelation than the flow magnitude.

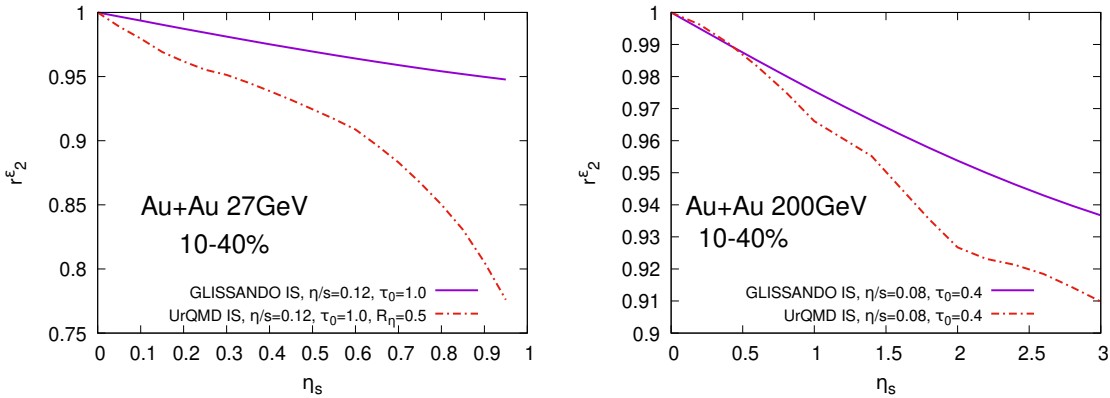

Figure 3: Space-time rapidity dependence of the initial state eccentricity decorrelation for $10-40\%$ Au-Au collisions at $\sqrt{s_{\mathrm{NN}}} = 27$ and 200 GeV. Figure taken with permission from [6].

# Acknowledgements

JC, IK, and BT acknowledge support by the project Centre of Advanced Applied Sciences, No. CZ.02.1.01/0.0/0.0/16-019/0000778, co-financed by the European Union. JC and BAT acknowledge support from from The Czech Science Foundation, grant number: GJ20-16256Y. IK acknwowledges support by the Ministry of Education, Youth and Sports of the Czech Republic under grant "International Mobility of Researchers – MSCA IF IV at CTU in Prague" No. CZ.02.2.69/0.0/0.0/20_079/0017983. BT acknowledges support from VEGA 1/0348/18. Computational resources were supplied by the project "e-Infrastruktura CZ" (e-INFRA LM2018140) provided within the program Projects of Large Research, Development and Innovations Infrastructures.

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
