# Peer review of "Flow decorrelation in heavy-ion collisions at $\sqrt{s_{_{\rm NN}}}$=27 and 200 GeV with 3D event-by-event viscous hydrodynamics"

_SciPost Physics Proceedings_

## Round 2 · Referee Report · Anonymous (Referee 1) · 2022-1-3

Strengths
- The paper is clearly written
- The paper attempts a clear explanation of a hot-topic issue
- The calculation is novel, to my knowledge it has not been performed before
Weaknesses
- It is unclear what the physics origin of the difference between the two employed initial state models is
Report
These proceedings presents a calculation of flow decorrelation in pseudo-rapidity, using state-of-the-art viscous hydro with two different initial state models. By repeating the same calculation with two different initial state model, the authors point out the importance of the initial state model in explaining elaborate flow-related quantities.
While the authors present (in fig. 3) a clear demonstration of the difference in initial state eccentricity decorrelation between the two initial state models (UrQMD and GLISSANDO), the initial state models are not explained to a degree that the readers understand why this difference comes about (are they not both Glauber models?) My only requested change relates to that.
While the authors present (in fig. 3) a clear demonstration of the difference in initial state eccentricity decorrelation between the two initial state models (UrQMD and GLISSANDO), the initial state models are not explained to a degree that the readers understand why this difference comes about (are they not both Glauber models?) My only requested change relates to that.
Requested changes
- I would like the authors to spend a couple of lines explaining why the initial state eccentricity decorrelation is so drastically different between the two models. I would naively have expected them to be almost identical, since I thought the two were both based on simple Glauber.

Author: Jakub Cimerman on 2022-02-07 [id 2163]
(in reply to Report 1 on 2022-01-03)We thank the referee for the report. Due to page limitations, we did not go into details describing the difference between models. However, to respond to the referee's request, we added a line to the UrQMD model description to emphasize the difference between the models. We also added a sentence to the interpretation of Fig. 3 describing the other effect causing the difference between the models - switching from hadron cascade to fluid.

---

## Round 3 · List of Changes

- added line in Section Model to emphasize the difference between initial state models UrQMD and Glissando
- added a sentence to the interpretation of Fig. 3 describing the other effect causing the difference between the models - switching from hadron cascade to fluid

You are currently on this page

Resubmission 2110.14783v3 on 8 February 2022

---

## Editorial Decision

publication_decision_taken:_accept